# Lymphadenectomy for Upper Tract Urothelial Carcinoma: A Systematic Review

**DOI:** 10.3390/jcm8081190

**Published:** 2019-08-08

**Authors:** Igor Duquesne, Idir Ouzaid, Yohann Loriot, Marco Moschini, Evanguelos Xylinas

**Affiliations:** 1Department of Urology, Cochin Hospital, Assistance Publique-Hôpitaux de Paris, Paris Descartes University, 75014 Paris, France; 2Department of Cancer Medicine, Gustave Roussy Institute, Cancer Campus, Grand Paris, University of Paris-Sud, 94800 Villejuif, France; 3Department of Urology, Bichat-Claude Bernard Hospital, Assistance Publique-Hôpitaux de Paris, Paris Diderot University, 75018 Paris, France; 4Klinik für Urologie, Luzerner Kantonsspital, 6004 Lucerne, Switzerland; 5Department of Urology, Bichat-Claude Bernard Hospital, Assistance Publique-Hôpitaux de Paris, Paris Descartes University, 75006 Paris, France

**Keywords:** lymphadenectomy, renal pelvis, ureter, upper tract, urothelial carcinoma, recurrence, survival, outcomes

## Abstract

Background: The role of lymphonodal dissection during surgery for a tumor of the urinary tract remains controversial. Objective: To analyze anatomical bases of lymphonodal dissection in tumors of the upper urinary tract and analyze its impact on survival, recurrence, and staging. Acquisition of data: A web-based search for scientific articles using Medline/Pubmed was carried out to identify and analyze articles on the practice and the role of lymphonodal dissection in this indication. Data Synthesis: The lymphatic drainage of the upper urinary tract has rarely been studied and is poorly understood. The lymphonodal metastatic extension is the most common extension in upper urinary tract urothelial carcinoma. Lymphnode invasion is a clear independent poor prognostic factor. Therefore, it seems legitimate to offer an extended lymphonodal dissection to patients undergoing surgery to cure these tumors. When lymphnodes dissection respects clear anatomical principles based on the location of the primary tumor and its extension, it improves both survival and recurrence rates. This result could be secondary to the treatment of subclinical metastatic disease. Conclusion: An extended lymphadenectomy during surgery for upper urinary tract urothelial carcinoma following strict anatomical pattern improves staging with a highly probable therapeutic benefit.

## 1. Introduction

Upper tract urothelial carcinoma (UTUC) represents a rare but highly morbid entity. It accounts for approximately 5% of all urothelial tumor [1]. Regional lymph nodes invasion is the most frequent metastatic evolution for UTUC as it concerns about 30% to 40% of patients with muscle invasive UTUC [2]. Nodal involvement represents an independent predictive factor of lower survival rates [3]. Lymphadenectomy (LND) is generally accepted as a necessary part of procedures to treat numbers of different malignancy including urological tumors. The current gold standard treatment of high-risk non metastatic UTUC remains radical nephroureterectomy (RNU) including the removal of the ipsilateral bladder cuff. Selected patients with low-risk disease (small size less than 2 cm, unifocal, well-differentiated tumors) could be treated conservatively with kidney-sparing strategies such as segmental ureterectomy [4].

Lymph node dissection is now accepted as a mandatory step of surgery for localized muscle-invasive bladder urothelial cancer management; therefore, investigators show growing interest for lymphadenectomy during the RNU. The realization of a lymph node dissection during RNU was first described in the 1970s [5]. Its indication, extent and anatomical templates according to tumor location are still under evaluation. Moreover, its therapeutic and staging benefits are still highly debated in the current literature.

In this systematic review, we aim to bring lightings on the latest important understanding of LND in UTUC management with a specific interest in anatomical aspects.

## 2. Materials and Methods

A systematic review of the literature was performed following the Preferred Reporting Items for Systematic Reviews and Meta-Analyses (PRISMA Statement) criteria [6]. A Pubmed/MEDLINE web-based search was conducted without time limit up to January 2019 using the following keywords: “Upper Tract Urothelial Carcinoma” and “Lymphadenectomy” or “Lymph Node Dissection” or “Lymph Node Excision.” The literature search was limited to English-language articles. This research resulted in a total of 157 articles. There were no randomized controlled trials. Articles were selected if they provided information on the lymph node dissection (LND) indication, a description of the surgical technique including anatomical template, or survival data according to the nodal status. We did not take into account the clinical nodal status (cN0 or cN+) to select the studies. Clinical case reports, other-languages articles, and articles dealing with different subjects were excluded. Abstracts were read by first author (Duquesne I). The article quality evaluation was then performed by three authors (Duquesne, I., Xylinas, E., and Ouzaid, I.). Article quality were assessed on clinical interest and on the quality of the described cohorts. A list of 34 relevant articles was selected and retrieved for further qualitative analysis. The detail of the selection of articles is shown in Figure 1.

For each eligible trial, the following data were collected, if available: study design, detailed descriptions of anatomical templates, number of studied patients, overall survival, specific survival rates, and recurrence-free survival.

## 3. Results—Evidence Synthesis

### 3.1. Indication of Lymph Nodes Dissection (LND) in the Surgical Management of UTUC

There are no existing guidelines on the indication of LND during surgical treatment of upper tract urothelial carcinoma (UTUC). The reviewed studies were mostly retrospective designs, preventing authors from defining standardized indication criteria for performing LND. A large majority of studies reported the indication as at “surgeon’s discretion” based on clinical presentation, location, and laterality of the primary tumors.

The most common criterion for lymph node dissection performance found in the analyzed studies was the risk of disease progression. Rajput et al. [7] performed LND in patients with high-risk disease defined as high grade on biopsy, large-volume tumor, or sessile architecture [8]. In a multicenter study published in 2009, Roscigno et al. [9] excluded from analysis patients with pTa disease and with pTis disease as they had a negligible risk of LN metastases. Selecting patients on preoperative assessment of lymph node status seems difficult as clinicians do not have available imaging, cytology or pathology techniques to assess lymph node invasion preoperatively in UTUC [10].

Divergent positions on the cN+ nature as indication for LND were found in the reviewed articles. The majority of authors [11,12] excluded cN+ patients, defined as having at least a suspicious lymph node of more than 1 cm on pre-operative imaging, arguing that pN+ patients may not benefit from lymphadenectomy and therefore extrapolating these thoughts to cN+ patients, thus referring cN+ patients to induction chemotherapy. In the small number of studies dealing with cN+, it seems hard to analyze whether the survival gain is secondary to LND or to systemic chemotherapy. Some authors chose to perform LND in patients with an infiltrative disease or with enlarged nodes on a preoperative evaluation (CT scan) and in cases of enlarged nodes discovered perioperatively [11].

The clearest information on the indications of LND was provided by a large retrospective study of the US Surveillance, Epidemiology, and End Results program (SEER) registry. This study showed that surgical teams performed more LND in patients diagnosed at an early age, in patients with stage T3–T4, in patients with larger tumors, or in patients with left sided tumors [13]. Despite the lack of clear recommendation in the international guidelines, there is an uptake in the proportion of LND performed based on the SEER cohort between 2004 and 2012 (20% vs. 33%).

### 3.2. Definition of Anatomical Templates

There is no existent comprehensive description of lymphovascular drainage of the upper urinary tract. The first studies describing the areas of nodal extensions of tumors of the upper urinary tract date from the 1980s. They demonstrated that the anatomical regions concerned by lymph node invasion in UTUC were para-aortic areas, latero-caval areas, and pelvic areas regarding tumor of the distal ureter [14]. These original principles helped to define the first TNM classification for this cancer. Table 1 displays the different definitions of anatomical templates depending on the location of the primary tumor when defined in the included studies.

The retrospective design of the majority of studies do not allow precise definitions, and in most cases, the anatomical templates were chosen at the surgeon’s discretion. The primary tumor location is usually divided into eight parts: the right renal pelvis, the right upper ureter, the right middle ureter, the right lower ureter, the left renal pelvis, the left upper ureter, the left middle ureter, and the left lower ureter. The upper ureter is defined as the upper third of the ureter, which is superior to the inferior mesenteric artery. The middle ureter is defined as the middle third of the ureter from the level of the inferior mesenteric artery to the crossing with the common iliac artery. The lower ureter is the distal third of the ureter below the crossing.

In a pioneer study from 2007, Kondo et al. [17] retrospectively examined the primary site and incidence of lymph node metastases of UTUC on imaging and pathological studies of surgically resected specimen. Authors studied 181 patients, including 42 patients with clinical nodal involvement preoperatively known. The incidence of nodal involvement was 20% to 30% for tumors of the renal pelvis, upper ureter, and middle ureter. Nodal involvement from tumors of the lower ureter (right or left) was less frequent, at about 10% compared with the other tumor locations. Among 15 patients with right renal pelvis tumor who had nodal involvement, a total of eight right renal hilar nodes were positive (53%), five positive paracaval nodes (33%), and five positive retrocaval nodes (33%). Among 15 patients with left renal pelvic cancer who had nodal involvement, nine positive nodes were located to the left renal hilar area (60%), eight positive nodes were located in the abdominal para aortic area, and one positive node was interaorticocaval. Only two patients had right upper ureter cancer with nodal involvement: one positive node was retrocaval, and one interaorticocaval. Three patients presented with right middle ureter with positive nodes—two positive nodes were in the interaorticocaval position (66%), and one was retrocaval (33%). Among patients with left middle ureter cancer, one had nodal involvement in the abdominal paraaortic area. Two patients had right lower ureter cancer with positive nodes: one common iliac node (50%), and one obturator node (50%). Two patients had left lower ureter cancer with nodal involvement: one node was found in the left common iliac zone (50%), and one positive node was left internal iliac (50%). Among other information, this study has shown that metastatic extension of tumors of the right pelvis and of the first two thirds of the right ureter had in fact a broader expanse than the one described previously.

Based on these preliminary data, Kondo et al. [21] retrospectively evaluated the impact of the extent of LND on survival. For this study, investigators used their previous results to define the anatomical template for LND. For tumors of the right renal pelvis and the right upper and middle ureter the renal hilar, paracaval, and retrocaval nodes (and interaorticocaval nodes for ureteral tumor) were considered regional lymph nodes to remove. For tumors of the left renal pelvis and the left upper and middle ureter, surgeons had to remove renal hilar and para aortic nodes “en bloc.” For tumors of the lower ureter, LND concerned bilateral pelvic nodes including common iliac, external iliac, obturator, and internal iliac nodes.

The first results developed by Kondo et al. in 2007 were confirmed at the time of updating its primary study in 2012 with a cohort of 77 patients [22]. Investigators then offer to clean out the areas where the incidence of metastases was statistically at least 10%. However, these findings must be tempered by the very low numbers of tumors studied at each site however they closely match the results obtained by Assouad et al. in their anatomical study [23]. Figure 2 schematically summarizes the anatomical templates to use depending on the location of the primary tumor.

The clear anatomical definition of the LND area seems beneficial from an oncological point of view. Indeed, in a study published in 2016, Furuse et al. showed that LND performance was better when the anatomical territory to dissect was well defined [12].

### 3.3. Staging Role of Lymph Node Dissection

Regarding UTUC, Roscigno et al. [24] have shown that lymphovascular invasion defined as the presence of tumor cells within an endothelium lined space without underlying muscular walls is an independent predictor of lower Disease Free Survival (DFS) and Cancer Specific Survival (CSS).

Table 2 displays patients’ outcomes in the different studies analyzed in our review according to nodal status. These data confirm the negative prognostic impact of lymph node involvement in patients harboring UTUC. Thus, a proper post-operative nodal status appears essential for an adequate management of patient and a good selection of patients who may benefit from adjuvant systemic therapy administration.

The discordant results of the studies presented, especially between studies reporting a benefit of lymphadenectomy for pN0 patients compared to pNx (no LND) and those only reporting the results of pN+ patients, can be explained by the difference in patient numbers between studies but also by the difference in lymphadenectomy extension in the different studies. A more extensive dissection allowing a more important profit for the patients classified as pN0. This concept, although intuitive, remains to be demonstrated.

The role of LND becomes clearer when the tumor stage is taken into account. Roscigno et al. [24] illustrated this idea by demonstrating how the improvement in five-year CSS between pN0, pNx, and pN+ patients was more important in the pT2 or higher population (70% vs. 58% vs. 33%; *p* = 0.017 and *p* < 0.01) rather in the pT1 or higher population (71% vs. 69% vs. 35%; *p* = 0.032 and *p* < 0.01). Similar results were found regarding the Recurrence Free Survival (RFS) for pT2 or higher tumors as shown in Abe et al. [19], where RFS was significantly higher in pN0 patients compared to pNx patients. These two studies highlight that LND has a bigger staging effect when performed in muscle invasive cancer population.

All reported studies in our review support a clear staging benefit when LND is performed.

### 3.4. Therapeutic Role of Lymph Node Dissection

The therapeutic role of lymph node dissection remains controversial even in the management of other urological malignancies including in the management of clear cell carcinoma type of kidney cancer and in prostate cancer in particular because of the low incidence of lymph node extension in these diseases.

In a retrospective multicenter study on 1130 patients from 2009, Roscigno et al. [24] demonstrated that there were no significant outcome differences between patients who underwent LND and patients who did not. Indeed, the five-year DFS was 60% for LND and 65% for no LND patients (*p* = 0.12), and the five-year CSS was 66% for LND patients versus 69% for no LND group (*p* = 0.23). In the same study, both DFS and CSS increased incrementally from pN+ to pNx to pN0 patients: 29% vs. 66% vs. 71% (*p* < 0.001 and *p* = 0.045 respectively) and 35% vs. 69% vs. 77% (*p* < 0.001 and *p* = 0.032 respectively). pN0 cases might beneficiate of the LND in having possible micro metastasis removed. Therefore, pN+ patients might benefit from systemic adjuvant chemotherapy administration.

Patients who did not undergo LND (pNx) presented an almost comparable outcome compared to pN0. That could be explained by selection bias, as LND was performed in a more severe population. Discordant results were obtained in 2011 by Burger et al. [30] in a retrospective international study of 785 patients. In this cohort, patients who did not receive any LND during NU for UTUC had worse cancer-related outcomes compared with those with pN0 cancer. However, this conclusion only applies to patient with locally advanced disease. Indeed, investigators proved in a subgroup multivariate analysis limited to patients with locally advanced disease that pN0 patients’ diseases have a significantly lower risk of recurrence (HR: 0.3; *p* < 0.001) and death (HR: 0.3; *p* < 0.001) when compared with pNx. Moreover, in univariate and multivariate analysis, patients presenting with lymphovascular invasion, pN0/pNx was not found to be predictor of either RFS (HR: 0.9; *p* = 0.585 for univariate and HR: 1.1; *p* = 0.631 for multivariate) or CSS (HR: 1.0; *p* = 0.962 for univariate and HR: 1.3; *p* = 0.223). The benefit of LND might be greater for larger localized tumors as reflected in the retrospective study published by Ikeda et al. where a clear benefit of LND for five-year DFS and five-year CSS was proved when performed in ≥pT3 patients [32].

The benefits of LND are less clear in cN0 patients. Its efficiency seems to depend on its technique of realization. In a prospective multicenter study from 2014, Kondo et al. [34] demonstrate the importance of using predefined anatomical templates in the management of cN0 patients. Indeed, both specific survival (89.8% vs. 51.7%, *p* = 0.01) and overall survival (86.1% vs. 48.0%, *p* = 0.01) were statistically improved in the lymph node dissection group. Similarly, progression-free survival of patients was almost significantly improved in the LND group (77.8% vs. 50.0%; *p* = 0.06). A retrospective study of 418 patients published by Yoo et al. found no significant difference in terms of disease recurrence and survival between patients undergoing LND and patients who did not [31]. These data evaluating the therapeutic role of LND are consistent with the finding that older age patients, a ≥cT3 stage disease and cN+ disease were found as preoperative predictors for pN+ disease by multivariate analysis in a large retrospective study [33].

The therapeutic role of lymphadenectomy remains difficult to prove, as indications for adjuvant systemic treatments such as chemotherapy was not precisely defined. Indeed, in the current studies, it appears difficult to distinguish gain in survival rates secondary to a multimodal management rather than lymph node dissection itself.

### 3.5. Number of Lymph Nodes Needed to be Removed

A parallelism was made between urothelial cancer of the upper urinary tract and infiltrative bladder tumors where the number of lymph nodes removed is a reflection of the quality of the dissection at the time of radical cystectomy [35]. Table 2 shows the median or mean number of lymph nodes removed in each study included in our review (when the data was available) and reveals how great the external variability is between each study as well as the internal variability inside each study between patients. This variability is mainly secondary to the lack of consensus on the surgical technique and on the anatomical pattern to use depending on the location of the primary tumor.

In 2009, Roscigno et al. [9] published in the European Association of Urology Journal an international multicenter study gathering 562 patients to determine the right number of lymph nodes to remove. The median number of removed LNs was five (range 1–41). The number of LNs removed was independently associated with recurrence (HR: 0.97; *p* = 0.04) but not with Cancer Specific Mortality (CSM) (*p* = 0.1) in univariate analysis. In multivariate analysis the number of LNs removed was associated with both recurrence and CSM (*p* < 0.02). In this same study, investigators demonstrated that the most informative cut-off for the number of LNs removed was eight for recurrence as well as for CSM. Indeed, the recurrence rate was lower in patients treated with an extended LND (≥8) (HR: 0.49; *p* < 0.01) as well as the CSM probability which decreased with extended LND (HR: 0.42; *p* < 0.01). The five-year survival of patients treated with an extended lymphadenectomy (≥8 LNs) was statistically significantly higher than in the patients treated with a limited lymphadenectomy (<8 LNs) (84% vs. 73% *p* = 0.038). The removal of eight nodes resulted in a 75% probability to detect at least one positive node [36]. These two studies strengthen preliminary results from Roscigno et al. published in 2008 [37], a monocenter study showing that an increasing number of LNs removed was associated with lower risk of DFS and of CSS (*p* = 0.006 and *p* = 0.009 respectively).

The studies of Roscigno et al. are confirmed by Kondo et al. in 2010 [34]. In a study of 80 patients with tumors at least pT2cN0, the authors fail to demonstrate that the number of nodes removed improves survival (*p* = 0.07), but they did demonstrate that patients with more than eight lymph nodes removed tended to have a specific benefit on survival. Of course, these studies are subject to bias on the technicality of the pathological analysis of lymph nodes, especially their exact account. The most important bias is the variability introduced by multiple surgeons performing interventions in the different studies. However, it would be interesting to consider this aspect as reflecting clinical practice. These limitations are to be weighted by what Dorin et al. show in a study published in 2011 [38]. The dissection, i.e., the surgical procedure, is more important than the account of lymph nodes by pathologists in terms of therapeutic outcome and staging.

More recently, Xylinas et al. developed a model that would allow clinicians to determine the probability that a patient classified as pN- by pathologists truly has no lymph node metastasis. This model was developed on 814 patients within an international multicenter cohort [39], and external validation was obtained on 2768 patients who underwent RNU with LND [40]. The median number of LNs removed was five (range 1–88 Interquartile Range (IQR) = 5). In this study, 66.5% of all patients were considered as pN0 according to final pathology reports. The authors have naturally confirmed that probability of missing lymph node metastasis decreased as the number of examined LNs increased. When only one LN was surgically removed, the authors estimated that 35% of patients actually pN+ would have been classified as pN0. This percentage drops to 8% when the number of LNs removed is five (median of the cohort), at 5% when the number of LNs removed is 8, and 4% when the number of LNs is 10. The larger the size of the tumor, the more it is necessary to remove a large number of LNs. Concerning pT3–4 tumors, at least 12 LNs are required for more than 95% of patients classified as pN0 to have no real lymph node metastasis. This model might encourage surgeons to perform extensive dissection to obtain a reliable lymph node staging.

### 3.6. Role of Lymph Node Density

As seen above, a minimal number of LNs should be dissected to beneficiate from staging or therapeutic role of LND in UTUC. Does LND become irrelevant when insufficient numbers (less than eight) of LNs are removed? As shown in Table 2, the majority of patients undergoing LND have a number of removed nodes inferior to eight. The number of lymph nodes removed does not seem to be a satisfactory indicator to evaluate the effect of LND on patients’ outcome. Some authors developed the lymph node density in UTUC.

Lymph node density is defined as the mean ratio of positive nodes to nodes removed. Bolenz et al. [41], in a multi-institutional cohort of 432 patients who underwent RNU with LND, studied the effect of lymph node density. They showed that a Lymph node Density ≥30% was associated with a greater five-year recurrence rate (25% vs. 38%; HR: 1,8; *p* = 0.021) and a greater five-year cancer-specific mortality rate (30% vs. 48%; HR: 1.7; *p* = 0.032). Mason et al. demonstrated in their Canadian retrospective trial of 2012 [29] that in the multivariable model, the ratio of positive nodes to nodes removed was associated with decreased DSS (HR: 2.70; *p* = 0.001), RFS (HR: 1.94; *p* = 0.015), and OS (HR: 2.34; *p* = 0.013) and with a cut-off point of ≤20%. Lymph node density does appear as a more precise index to predict outcome after lymph node dissection.

### 3.7. Impact of Surgical Approach on Lymph Node Dissection

RNU with ipsilateral bladder cuff excision can be performed either with open techniques or laparoscopically. Laparoscopic techniques include pure laparoscopic surgery, robot-assisted surgery, and a hand-assisted approach. The LND itself can be performed during a laparoscopic Nephroureterectomy (NU) either laparoscopically or through an open approach (Pfannenstiel or Gibson incision).

In 2016, Pearce et al. [42] published a clinical study assessing the effect of surgical approach on performance of LND and perioperative morbidity for radical NU. This prospective multicenter report used the US National Inpatient Sample between 2009 and 2012 to identify 16619 patients who underwent NU. Fifteen percent of all patients underwent procedure with LND. Patients undergoing Robotic NU were more likely to undergo LND (27%) when compared to open surgical approach (15%) and laparoscopic approach (10%) (*p* < 0.001). Regarding LND performance during NU, a robotic approach increased the odds of LND (OR = 1.9; 95% CI: (0.3–1.8); *p* = 0.001), whereas a laparoscopic approach decreased the odds of LND (OR = 0.6; 95% CI: (0.4–0.8); *p* = 0.004). Roscigno et al. [9] demonstrated in their study that there were no statistical significant differences between the median number of removed lymph nodes when LND was performed during open surgery or laparoscopically: five (1–41) versus four (1–21) *p* = 0.12. Logically, there was no significant effect of the surgical approach on recurrence (*p* = 0.1) or mortality (*p* = 0.1). Similar results are reported by Abe et al. [43] on retrospective data from three Japanese academic centers gathering 214 patients cTanyN0M0. They reported a comparable median LN number between the open group and the pure laparoscopic group (12 vs. 11.5). The large median number of LNs removed compared to the majority of published studies shows that these studies are carried out in expert centers and therefore lose external validity.

This center effect is well illustrated by several studies reporting contradictory results. Two large US studies using data from the National Cancer Database between 2010 and 2013 [44,45]. A higher proportion of LND performed in patients operated by open or robot-assisted approaches than by a pure laparoscopic approach is noted (OR = 0.77 *p* <0.01). In addition, when performed, laparoscopically LND seems less effective since the number of lymph nodes removed is significantly lower than by open or robot-assisted surgery. The benefit of robotic assistance is confirmed by a monocentric retrospective study by Melquist et al. [20]. The number of LNs removed was significantly greater in the robot-assisted arm than in the pure laparoscopic arm (21.0 (IQR 16.0–30.0) vs. 11.0 (IQR 5.5–21.0, *p* < 0.0001).

Regarding intraoperative complications Pearce et al. demonstrated that there was no significant difference in the rate of events by surgical approach (*p* = 0.1). The rate of any postoperative complication was lowest for robotic NU (19%) and highest for open NU (30%) (*p* < 0.001). The robotic assisted approach was also safer with a lower risk of transfusion (8% vs. 30% *p* = 0.012), despite a longer operating time (5.1 vs. 3.9 h, *p* = 0.0001). There were no other differences in postoperative outcomes apart from a longer stay of one day in the robot group. The good results of the robot-assisted approach are to be measured because we note that patients operated by robot are mostly younger and with smaller tumors than patients operated on using other techniques [44].

Concerning the oncological results, the surgical approach does not seem to make any difference in specialized centers. In the Japanese multicenter study from Abe et al. [43], comparable oncological results between open and laparoscopic groups including similar five-year RFS (71.7 vs. 74% (*p* = 0.7829)), similar CSS (77.8% vs. 80% (*p* = 0.8441)), and similar OS (72.8% vs. 75.9% (*p* = 0.3456)). There was no difference between the two approaches for pT3–4 tumors. The same conclusions were drawn by Kido et al. in a monocenter study comparing laparoscopic and open approaches [46]. However, these results must be put into perspective because laparoscopic patients are often selected for their good general condition and often have smaller tumors.

### 3.8. Safety of Lymph Node Dissection

A large multicenter retrospective study relating to a population of 16619 patients who underwent NU for urothelial carcinoma with LND (*n* = 14059; 85%) or without LND (*n* = 2560; 15%) assessed the surgical approach on regional LND performance and complications. They demonstrated that LND increased the risk of complications (OR = 1.3; 95% CI: (1.001–1.7); *p* = 0.049) [42]. Only one prospective study specifically focused on LND complications. This monocenter prospective single-arm trial included 20 patients with a mean follow-up of 12 months (2–24) and with a mean total of lymph nodes removed of 7 (IC 95% [2,3,4,5,6,7,8,9,10,11,12,13,14,15,16,17]). Of the 20 patients, eight presented with minor postoperative complications (defined as Clavien Grade I–II), and only one patient had a major complication, which consisted of a chylous leak requiring secondary surgical lymphostasis (Clavien IIIb) [11]. In the other studies reviewed, no further complications specifically related to LND were reported.

## 4. Key Concepts

−Lymph node dissection performance tends to increase;−Lymph node dissection is usually performed for high risk UTUC;−Lymph node dissection might benefit for ≥T3 and high-grade patients both in cN0 and cN+ patients;−Lymph node dissection should follow a strict anatomical template depending on the location of primitive tumor;−Performing LND is safe and does not increase surgical complications;−Surgical approach does not seem to have major influence on the LND performance.;−LND might improve staging;−LND should remove eight lymphnodes;−Lymph node density might be a more precise index to predict outcome than lymph node count.

## 5. Conclusions

The recommendations that we can draw from this review would be grade III. This review suggests that lymphonodal dissection during surgery of tumors of the upper urinary tract improve staging and can help selecting candidates for systemic adjuvant therapy. This study also shows a probable therapeutic role of lymph node dissection with an interest in the survival of pN0 patients compared to pNx subgroup (treatment of subclinical metastases) and interest in pN+ patients (treatment of lymph node invasion).

To enable these benefits, achieving lymphadenectomy must follow a number of principles that this review presents. The lymphadenectomy should strictly follow predefined anatomical patterns that are adapted to the location of the primary tumor. The lymphadenectomy should be broad and remove a minimum of eight lymph nodes. If the number of removed lymph nodes is less than eight, one must evaluate the lymph node density.

This review supports the realization of a lymphonodal dissection during surgery for urothelial tumors of the upper urinary tract. The results presented are mostly from trials with a low level of evidence, so it is necessary to conduct multicenter, prospective studies to further evaluate the practice of lymph node dissection.

## Figures and Tables

**Figure 1 jcm-08-01190-f001:**
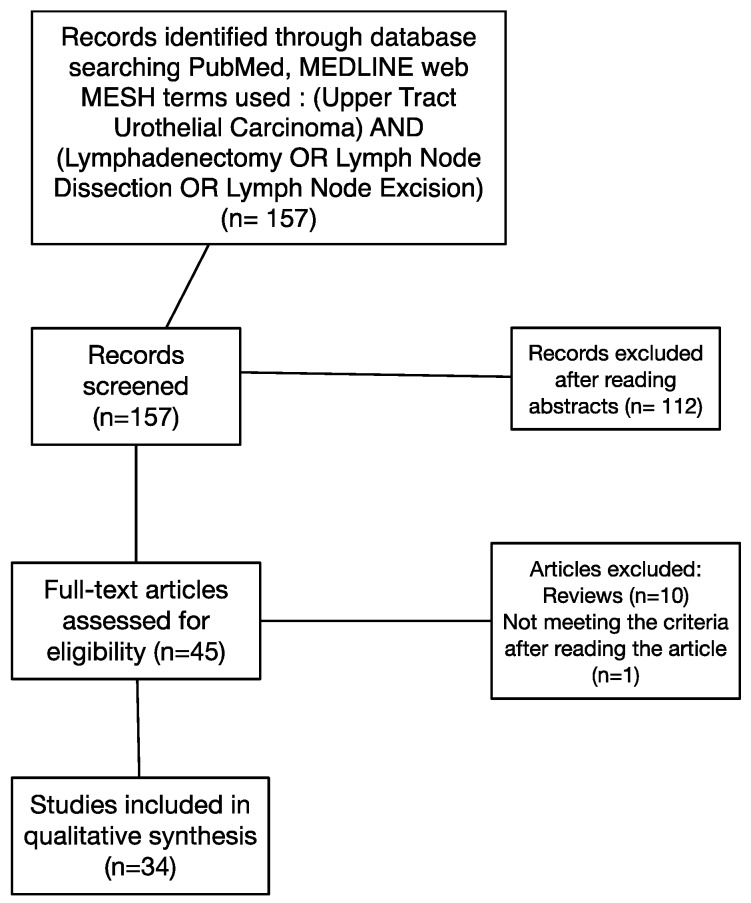
Flow chart.

**Figure 2 jcm-08-01190-f002:**
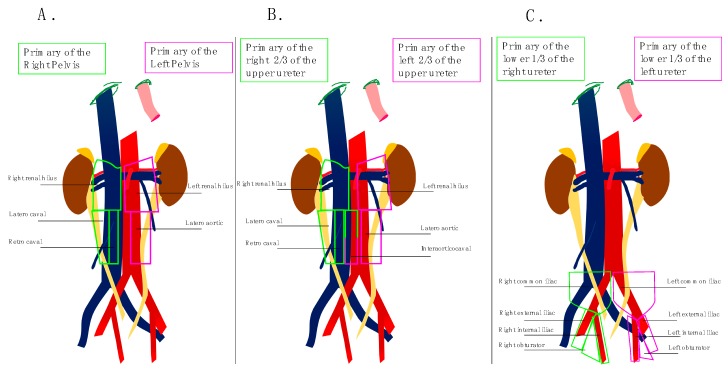
Lymph nodes anatomical templates based on primary tumors. (**A**) Primary tumor of the renal pelvis; (**B**) Primary tumor of the 2/3 of the upper ureter; (**C**) Primary tumor of the lower 1/3 of the ureter.

**Table 1 jcm-08-01190-t001:** Description of anatomical templates.

Study (Year)	Right				Left			
	Renal Pelvis	Upper Ureter	Middle Ureter	Lower Ureter	Renal Pelvis	Upper Ureter	Middle Ureter	Lower Ureter
Komatsu et al. [15] (1997)	Right side from the midline of the anterior surface of the aorta between the renal hilus and the aortic bifurcation		Right side from the midline of the anterior surface of the aorta between the renal hilus and the bifurcation of the common iliac artery	Right common iliac, external iliac, internal iliac, and obturator nodes	Left side from the midline of the anterior surface of the aorta between the renal hilus and the aortic bifurcation	Left side from the midline of the anterior surface of the aorta between the renal hilus and the bifurcation of the common iliac artery		Left common iliac, external iliac, internal iliac, and obturator nodes
Miyake et al. [16] (1998)	From the vena cava, between the renal hilus and the inferior mesenteric artery		From the vena-cava, between the renal hilus and bifurcation of the common iliac artery	Right pelvic nodes	From the para-aorta, between the renal hilus and the inferior mesenteric artery	From the para-aorta, between the renal hilus and bifurcation of the common iliac artery		Left pelvic nodes
Kondo et al. [17] (2007)	Right renal hilar, paracaval, and retrocaval nodes		Right renal hilar, paracaval, retrocaval nodes, and interaorticocaval nodes	Right common iliac, external iliac, obturator and internal iliac nodes	Left renal hilar, paracaval and retrocaval nodes	Left renal hilar, para-aortic nodes		Left common iliac, external iliac, obturator and internal iliac nodes
Brausi et al. [18] (2007)	Para-aortic, paracaval, or interaortocaval nodes from the renal hilus to the inferior mesenteric artery		Para-aortic, paracaval, or interaortocaval nodes from the renal hilus to the common iliac artery	Right pelvic nodes	Para-aortic, paracaval, or interaortocaval nodes from the renal hilus to the common iliac artery	Para-aortic, paracaval, or interaortocaval nodes from the renal hilus to the common iliac artery		Left pelvic nodes
Rajput et al. [7] (2011)	Retroperitoneal LND		Right pelvic nodes		Retroperitoneal LND	Left pelvic nodes		
Rao et al. [11] (2012)	Right perihilar lymph nodes, paracaval lymph nodes, right pelvic lymph nodes (common external and obturator lymph nodes). Removal of interaortocaval nodes was left to the discretion of the surgeon, depending on the presence of positive paracaval nodes as determined preoperatively or on intra-operative frozen section.				Left perihilar lymph nodes, para aortic lymph nodes, left pelvic lymph nodes (common external and obturator lymph nodes). Removal of interaortocaval nodes was left to the discretion of the surgeon, depending on the presence of positive para aortic nodes as determined preoperatively or on intra-operative frozen section.			
Matin et al. [8] (2015)	Right hilum to vena cava bifurcation, including paracaval (including precaval region) and retrocaval nodes. Additional dissection of interaorticocaval and common iliac nodes was performed when suspicious nodes were identified in these regions on preoperative imaging or upon visual inspection intraoperatively.		Para-aortic in addition to right common and external iliac nodes. Additional paracaval or para-aortic was performed based on imaging intraoperative inspection or surgeon discretion.	Right pelvic lymphadenectomy (common, external, internal, and obturator). Additional paracaval or para-aortic was performed based on imaging intraoperative inspection or surgeon discretion.	Left hilum to origin of inferior mesenteric artery, including para-aortic nodes (including preaortic nodes). Additional dissection of interaorticocaval and common iliac nodes was performed when suspicious nodes were identified in these regions on preoperative imaging or upon visual inspection intraoperatively.	Para-aortic in addition to left common and external iliac nodes. Additional paracaval or para-aortic was performed based on imaging intraoperative inspection or surgeon discretion.		Right pelvic lymphadenectomy (common, external, internal, and obturator). Additional paracaval or para-aortic was performed based on imaging intraoperative inspection or surgeon discretion.
Abe et al. [19] (2015)	Right renal hilar, paracaval, retrocaval plus interaortocaval			Right obturator, common iliac, external iliac plus internal iliac	Left renal hilar plus para-aortic			Left obturator, common iliac, external iliac plus internal iliac
Melquist et al. [20] (2016)	Hilar and precaval-paracaval-retrocaval regions plus interaortocaval dissection when technically possible.				Hilar with preaortic-paraaortic-retroaortic tissues plus interaortocaval dissection when technically possible.			
Furuse et al. [12] (2016)	Renal hilum, paracaval, retrocaval (including interaortocaval whenever possible)		Renal hilum, common iliac, paracaval, retrocaval (including interaortocaval whenever possible)	Common-external-internal iliac, obturator	Renal hilum, para-aortic	Renal hilum, common iliac, para-aortic		Common-external-internal iliac, obturator

**Table 2 jcm-08-01190-t002:** Oncological outcomes based on nodal status. The data in bold are statistically significant.

Study	Year	Study Interval	Number of Patients	Nodal Status (N° of Patients)	2-yr CSS, %	5-yr CSS, %	CSS: pN0 vs. pNx pN0 vs. pN+ pNx vs. pN+	2-yr DFS, %	5-yr DFS, %	DFS: pN0 vs. pNx pN0 vs. pN+ pNx vs. pN+	Median Number of Removed Nodes (IQ)	Median Follow-Up in Months (Range)
Komatsu et al. [15]	1997	1985–1993	36	pN0 (25) PN+ (11)	-	100 21	- -	- -	- -	- -	-	55 (3–135)
Miyake et al. [16]	1998	1986–1995	72	pN0 (22) pNx (37) pN+(13)	-	64 50 0	- - -	- - -	- - -	- - -	-	-
Brown et al. [25]	2006	1986–2004	184	pN0 (105) pNx (119) pN+ (28)	-	80 77 35	*p* = 0.58 - -	- - -	- - -	*p* = 0.85 - -	-	-
Kondo et al. [17]	2007	1989–2005	181	pN0 (139) pNx/PN+ (32/10)	- -	85.2 15.5	- -	- -	- -	- -	6 (2–30)	-
Brausi et al. [18]	2007	1980–2002	82	pN0/pN+ (24/16) pNx (42)	81.6 44.8	- -	- -	64.3 46.3	- -	- -	-	-
Secin et al. [10]	2007	1985–2004	255	pN0 (105) pNx (119) PN+ (28)	- - -	56 73 0	- - -	- - -	- - -	- - -	4 (2–10)	37 (-)
Novara et al. [26]	2007	1989–2005	269	pN0 (242) PN+ (27)	- -	82 12	- -	- -	- -	- -	-	-
Roscigno et al. [9]	2008	1986–2003	132	pN0 (69) pNx (37) PN+ (26)	- - -	73 48 39	***p*****= 0.001** - *p* = 0.476	- - -	72 39 35	***p*** **= 0.001** ***p*** **= 0.001** ***p*** **= 0.001**	8 (2–24)	42 (2–191)
Cho et al. [27]	2008	1986–2005	152	pN0 (54) pNx (89) PN+ (9)	- - -	72 67 63	*p* > 0.05 - -	- - -	91 80 71	HR 2.45 (0.26–22.47) **HR 3.91 (1.35–11.32)** -	6 (1–35)	-
Roscigno et al. [24]	2009	1987–2007	1130	pN0 (412) pNx (578) PN+ (140)	- - -	77 69 35	***p*****= 0.024** - ***p*****< 0.001**	- - -	71 66 29	***p*****= 0.045** - ***p*****< 0.001**	-	45 (1–250)
Lughezzani et al. [28]	2010	1988–2004	2842	pN0 (1835) pNx (747) PN+ (242)	- - -	81.2 77.8 34.2	*p* = 0.09 ***p*** **< 0.001** ***p*** **< 0.001**	- - -	- - -	- - -	-	43 (1–203)
Mason et al. [29]	2011	1990–2010	1029	pN0 (199) pNx (753) PN+ (77)	- - -	72.1 74.7 29.8	HR 0.96 (0.64–1.44) **HR 2.97 (1.47–6.01)** **HR 2.70 (1.56–4.69)**	– - -	39 41 7	HR 1.23 (0.78–1.96) **HR 2.94 (1.32–6.55)** **HR 2.83 (1.54–5.18)**	Mean: 4,3	19.8 (7.2–53.8)
Burger et al. [30]	2011	1987–2008	785	pN0 (136) pNx (595) pN+ (54)	- - -	79 77.4 26.7	*p* = 0.945 ***p*** **< 0.001** -	- - -	71.6 76.9 21.3	*p* = 0.586 ***p*** **< 0.001** -	3 (2–6)	34 (15–65)
Yoo et al. [31]	2016	1998–2012	418	pN0 (116) pNx (286) pN+ (16)	- - -	OS = 80.2 OS = 71.7 OS = 12.5	*p* = 0.230 - -	- - -	76.4 73.4 93.7	*p* = 0.682 - -	7 (3–10)	69 (-)
Ikeda et al. [32]	2017	1985–2013	404	pN0 (182) pNx (177) pN+ (40)	- - -	84.5 73.3 43.6	***p*****< 0.001** ***p*****< 0.001** -	- - -	78.3 61.9 33.2	***p*** **= 0.001** ***p*** **< 0.001** **-**	6 (3–10)	43 (17–89)
Inokuchi et al. [33]	2017	1995–2009	2037	pN0 (955) pNx (859) pN+ (223)	- - -	OS = 69.3 OS = 60.5 OS = 30	HR 1.03 (0.83–1.27) **HR 5.67 (4.56–7.05)** -	- - -	- - -	- - -	6 (3–11)	45.8 (21.8–75.9)

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
