# Peer review of "Lymphadenectomy for Upper Tract Urothelial Carcinoma: A Systematic Review"

_jcm, 2019, doi:10.3390/jcm8081190_

Round 1

Reviewer 1 Report

A nice systematic review on the roles of LND during RNU for UTUC.

The authors may want to address the following points prior to publication.

1) L116: "stage G3/G4" would need correction.

2) Table1, the top row of the "left lower ureter" column: right should be corrected to left.

3) L298: there seems to be a typographic error.

4) L321: bladder cuff may be better than cup.

Author Response

A nice systematic review on the roles of LND during RNU for UTUC.

We thank the reviewer for his kind comments and the time he spent on reviewing our manuscript. We hope to have displayed the current evidence on lymph node dissection for upper tract urothelial carcinoma.

The authors may want to address the following points prior to publication.

1) L116: "stage G3/G4" would need correction.

2) Table1, the top row of the "left lower ureter" column: right should be corrected to left.

3) L298: there seems to be a typographic error.

4) L321: bladder cuff may be better than cup.

All these were addressed in the revised manuscript.

Reviewer 2 Report

The authors summarized various aspects in terms of LND for upper urothelial carcinoma.

1.       Material and methods.

The authors should describe the details of selection process of relevant studies.  Who reviewed the abstracts of potential studies?  How many reviewers did they have before they selected the 34 relevant papers?  What did they do when opinions differed among the reviewers during the selection process ?

2.       Tables 1 and 2.  Reference number should be added to all the studies they summarized in the tables.  In addition, the authors did not describe the reference number in the sentences, such as lines 203-209 (Roscigno et al. and Abe et al), lines 241-246 (Kondo et al.), and lines 358-362 (Abe et al.).  Corresponding reference number should be added when they summarized the previous observations from different researchers.

3.       Figure 2 should be revised.  Not clear, almost graffiti.

Author Response

We thank the reviewer for his kind comments and the time he spent on reviewing our manuscript. We hope to have displayed the current evidence on lymph node dissection for upper tract urothelial carcinoma.

1.       Material and methods.

The authors should describe the details of selection process of relevant studies.  Who reviewed the abstracts of potential studies?  How many reviewers did they have before they selected the 34 relevant papers?  What did they do when opinions differed among the reviewers during the selection process ?

Two reviewers retrieved the abstracts and articles. When discordance appeared a third reviewer performed the final selection. This appears clearly now in the revised manuscript.

2.       Tables 1 and 2.  Reference number should be added to all the studies they summarized in the tables.  In addition, the authors did not describe the reference number in the sentences, such as lines 203-209 (Roscigno et al. and Abe et al), lines 241-246 (Kondo et al.), and lines 358-362 (Abe et al.).  Corresponding reference number should be added when they summarized the previous observations from different researchers.

References were added as asked.

3.       Figure 2 should be revised.  Not clear, almost graffiti.

Figure 2 is now of better quality.